# Direct Capture and Early Detection of Lyme Disease Spirochete in Skin with a Microneedle Patch

**DOI:** 10.3390/bios12100819

**Published:** 2022-10-02

**Authors:** Emily Kight, Rosana Alfaro, Shiva Kumar Goud Gadila, Shuang Chang, David Evans, Monica Embers, Frederick Haselton

**Affiliations:** 1Biomedical Engineering, Vanderbilt University, Nashville, TN 37211, USA; 2Division of Immunology, Tulane National Primate Research Center, Tulane University Health Sciences, Covington, LA 70433, USA

**Keywords:** Lyme disease, early detection, point-of-care, skin diagnostic

## Abstract

*Borrelia burgdorferi* sensu lato family of spirochetes causes Lyme disease (LD) in animals and humans. As geographic territory of ticks expands across the globe, surveillance measures are needed to measure transmission rates and provide early risk testing of suspected bites. The current standard testing of LD uses an indirect two-step serological assay that detects host immune reactivity. Early detection remains a challenge because the host antibody response develops several weeks after infection. A microneedle (MN) device was developed to sample interstitial fluid (ISF) and capture spirochetes directly from skin. After sampling, the MN patch is easily dissolved in water or TE buffer, and the presence of spirochete DNA is detected by PCR. Performance was tested by spiking porcine ear skin with inactivated *Borrelia burgdorferi*, which had an approximate recovery of 80% of spirochetes. With further development, this simple direct PCR method could be a transformative approach for early detection of the causative agent of Lyme disease and enable rapid treatment to patients when infection is early, and numbers of systemic spirochetes are low.

## 1. Introduction

Tick-borne diseases are major public and animal health issues worldwide. Ticks may transmit diseases such as babesiosis, anaplasmosis, ehrlichiosis, Lyme disease, Crimean Congo hemorrhagic fever, Powassan virus, and Rocky Mountain spotted fever to humans and animals [1]. According to the Centers for Disease Control and Prevention (CDC), the number of reported cases of all tick-borne disease has more than doubled over the past 13 years [2]. Of particular interest is Lyme disease (LD), the most common vector-borne illness in both North America and Europe, with an estimated incidence of ~476,000 cases per year in America alone [3]. Lyme borreliosis disease is a multisystemic infectious disease caused by a spiral-shaped bacterium, *Borrelia burgdorferi*, also referred to as spirochetes, that is transmitted to humans by the bite of infected species of *Ixodes* ticks [4]. Recent studies suggest LD will expand due to climate change making the more northern areas more temperate [5,6,7].

LD infection typically begins with erythema migrans (EM), an expanding skin lesion at the site of the tick bite. If left untreated, spirochetes may disseminate from the bite site leading to cardiac, rheumatologic and/or neurologic manifestations [8,9]. Conventional diagnosis of LD is often based solely on the presence of EM; however, it develops in only 70–80% of patients [10]. In the absence of EM, the signs and symptoms of Lyme disease are not specific enough to be clinically useful for diagnosis [11]. Additionally, misdiagnosis may occur due to the high degree of variability of skin lesions and the variation of interpretation [12]. Additionally, patients with darker skin often do not recognize early disease, resulting in increased rates of late manifestations [13].

In regions where tick-borne infections are common, any tick bite may be of concern. One strategy to assess exposure of tick-borne disease is direct testing of a removed tick. Tick testing centers offer polymerase chain reaction (PCR) testing of ticks, but these tests do not inform doctors if any tick-borne disease was transmitted to the patient’s body. As ticks fall off their host, many patients do not have the tick that bit them, so they are unable to use these testing services. Additionally, testing ticks requires a series of DNA isolation steps prior to PCR testing. During these several DNA isolation steps, small amounts of bacteria may not be detectable given that many commercial DNA extraction kits lose 80–90% of the starting material [14].

Currently, the CDC approved diagnosis procedure is a two-tiered testing method based on spirochete host response using an enzyme-linked immunosorbent assay (ELISA) first, and if the results are positive, another specificity test by Western blot is conducted. This two-tiered testing features high diagnostic specificity (95–100%) with high sensitivity (80–90%) for late stages of Lyme disease [15,16]. However, since development of host-antibody response takes several weeks, early diagnosis remains problematic. The 2-tiered testing suffers from just 40% sensitivity for early-stage (meaning that it misses more than half of the cases). It also requires multiple tests and is technically laborious leading to high costs.

After being deposited into skin, *B. burgdorferi* multiplies locally before migrating through tissues [4]. There are two main laboratory methods available to detect *B. burgdorferi* in human body fluids: in vitro culture of intact living bacteria, and PCR of a body fluid sample to detect bacterial DNA [17,18]. Though the culture method provides confirmation of active infection, the bacteria grow very slowly for up to 12 weeks in culture before reaching diagnostic concentration [2]. Cultures as a detection method are too time consuming and not practical for routine use. PCR has been used to amplify and detect *B. burgdorferi* DNA in urine, blood, and cerebrospinal fluid (CSF) [19]. However, PCR is not routinely used for these sample types [11] because spirochetes are present in low copy number which results in reduced PCR performance with these sample types [17]. Skin biopsies have also been used for DNA extraction methods with quantitative PCR detection [18,20] and was found to contain on average, 3381 + 544 spirochetes from a 2 mm skin sample [18]. In 2020, one study found that in 75.4% of patients who presented with EM, researchers could detect one or more *B. burgdorferi* genotypes in the skin using PCR [21]. However, obtaining a suitable skin biopsy may be painful as well as suffer from lower assay performance with PCR due to other inhibiting substances commonly used in tissue fixatives [22]. Additionally, a skin biopsy requires a trained nurse or doctor and the use of an anesthetic to numb the skin.

An inexpensive and simple means to collect the bacteria at the site where they are most concentrated may allow for early detection by PCR. Microneedles offer safe, affordable, and painless collection at the tick bite that may overcome the low numbers of spirochetes found in other sample types. The skin is the initial site of replication and persistence of *B. burgdorferi* [23]. During a tick blood-meal, *B. burgdorferi* does not migrate away from the feeding site until several days after the tick has detached from the host [24]. One study found detectable levels of bacteria remained in the skin for over a month in dogs [25]. This indicates that once the spirochetes are delivered into the skin, the spirochetes will multiply locally for several days and may still remain in the skin for longer periods of time. Therefore, it is likely that the bacteria load is concentrated in the skin and thus, easier to detect.

Since skin is easily accessed, sampling interstitial fluid (ISF) is an attractive alternative to blood but has received less diagnostic focus due to lack of methods for sampling [26]. Recently, studies have reported promising methods using MN to sample ISF [27,28,29]. Several types of MN have been designed to improve skin analyte sampling such as solid MN, hollow MN [30], porous MN [31], vacuum assisted MN [26], and hydrogel MNs [32,33]. Many recent MN studies have used poly(vinyl alcohol) (PVA) since it is affordable, exhibits low toxicity, and is a highly biocompatible polymer [34,35]. Another important property of PVA for sample collection is its swelling when immersed into a liquid sample driven by capillary force [36,37] which can result in absorbing up to 30% of its weight in a few minutes [38] and up to 450% of its weight when crosslinked with polyvinylpyrrolidone [39]. Additionally, PVA does not inhibit PCR [40,41] which allows downstream detection by PCR feasible. Crosslinked PVA hydrogels can be easily prepared by dehydration at room temperature without the use of any other chemicals [42]. These properties suggest that microneedles made from PVA are a promising approach for non-invasive detection of bacteria in a skin lesion.

In this work, we hypothesize that a microneedle patch can be used to capture spirochetes in skin and, with subsequent direct PCR of the collected sample, detect the causative bacteria for LD as shown in Figure 1. The feasibility of this approach is demonstrated with tick bite-appropriate PVA microneedle designs using in vitro and in vivo model systems.

## 2. Materials and Methods

### 2.1. Culture of a B. burgdorferi Strain B31

Low passage *B. burgdorferi* wild-type strain B31 clone 5A19 was grown in BSK-II supplemented with 6% rabbit serum (Pel-Freez Biologicals, Rogers, AR, USA) at 34 °C in a trigas incubator (5% CO_2_, 3% air, and the remainder N_2_). In addition to 6% rabbit serum, this media contained antibiotics rifampicin, 45.4 μg/mL; phosphomycin, 194 μg/mL; and amphotericin B, 0.25 μg/mL. Spirochetes were grown to late logarithmic phase and then pelleted at 2000× *g* for 30 min before inactivation. At the end of the run, the rotor was left to coast to minimize damage to the live spirochetes. The culture was washed twice using PBS and was inactivated by resuspension in 4% paraformaldehyde (PFA). Next, the tube was centrifuged at 2000 RPM for 15 min and resuspended in 1× PBS with 0.01% sodium azide for storage at 4 °C until use. SEM of the bacteria are shown in Appendix A.

### 2.2. Design of Microneedles

The tick hypostome (mouth) has a range of sizes [43]. The length of the hypostome varies depending upon sex and stage (nymph vs. adult) [44]. The adult female deer tick exhibits the largest hypostome which is ~500 μm long [45] and up to ~200 μm wide, while the males and nymphs have much smaller hypostomes [46]. Since the bacteria travel into a new host via tick saliva, the bacteria are likely deposited approximately close to the depth of the tick mouth. Researchers have used microscopy methods to image *Ixodes scapularis* feeding in the skin and observed a restructuring of the skin into a cone-shaped area surrounding the hypostome after 72 h of feeding, which could be filled with ISF [47]. In the skin, ISF is located beneath the stratum corneum (SC) which is ~10–200 µm thick [29]. To include the full range of tick hypostome length sizes and penetrate to a depth beyond 200 µm, we designed microneedle patches to be 300–800 µm in height, and 250 to 350 µm in base diameter to increase stability. Specifically, the center-to-center spacing has been shown to be a crucial factor in the penetration performance into skin [48], with more densely packed microneedles leading to the “bed of nails” effect [49]. Therefore, we selected a spacing of 1000 µm for our MN patch since this spacing has been studied with our range of heights and diameters in pig skin penetration tests [48]. Four microneedle cast print designs with different heights were created using Solidworks 2018.

### 2.3. Fabrication and Imaging of PVA Microneedles

Firstly, we developed a method to prepare PVA microneedle patches. Different types of PVA were tested ranging from 13–124 kDa molecular weight and 87–99% hydrolysis. Since a high viscosity solution cannot be pipetted with consistency for PCR, we tested the viscosity of the three molecular weight PVAs (Appendix A). We pipetted 50 µL of the three different PVA’s onto a weigh boat on a scale four times to examine the pipetting ability across the different molecular weight PVAs (Appendix A). The PVA *M*_w_ 85–124 kDa exhibited higher standard deviation between the pipetting and lower mass when pipetted. Additionally, for PVA *M*_w_ 85–124 kDa all four drops showed bubbles and the pipette tip exhibited bubbles as well which indicates the solution is too viscous for accurate pipetting. Therefore, we did not choose higher molecular weight PVA for further testing. We pipetted both PVA *M*_w_ 30–50 kDa and PVA *M*_w_ 13–23 kDa into microneedles silicone casts as both solutions were easily pipetted. However, the lower molecular weight PVA did not form rigid uniform microneedles in the same duration (one week under vacuum) as the other PVA, so the low molecular weight was not chosen for further testing.

The various types of PVA were cast and the PVA MNs that were solid and dry after a week were compared on a light microscope for needle shape and consistency. PVA with 30–50 kDa molecular weight and 98–99%-degree hydrolysis (Sigma-Aldrich, St. Louis MO, USA, cat# 363138-25G) was selected for further development. Cone-shaped microneedles were created for casting with dimensions 1000–1600 µm height, base 250–350 µm, and spacing 1000 µm apart were created with a 3D printer (ProJet^®^ 3510 HD plus, 3D Systems, Rock Hill, SC, USA) (Table 1). The method of fabrication PVA MN is shown in Figure 2. The 3D printed MNs were used to create a negative mold in silicone with a needle array of 5 × 5 cone shapes microneedles. PVA microneedles were then fabricated using the micromolding technique from a silicone mold (Mold Star**™** 30 silicone rubber, Smooth-on, Macungie, PA, USA). The 3D printed mold was cast in the silicone mold for 24 h to produce a negative mold under vacuum pressure of 100 PSI. The 3D print was then removed from the silicone mold. To make a 10% (*w*/*v*) PVA solution, PVA was dissolved in 10 mL deionized (DI) water at 90 °C on a hot plate with stirring for 12–24 h. To cast the final MN patch, PVA solution was pipetted into the silicone mold and then placed inside a vacuum at 100 PSI for 5–6 days to allow the formation of solid PVA microneedles.

### 2.4. Penetration Tests

Ideally, microneedles should penetrate the stratum corneum of the skin to allow of extraction of ISF. Additionally, if the microneedles do not penetrate beyond 500 µm, this enables painless collection. To investigate the penetration performance, various lengths of microneedle arrays were created and tested with the 3D print casts shown in Table 1.

To verify that the microneedles can penetrate the depth required, we tested the penetration of the microneedles in skin substitutes such as Parafilm M^®^ (PF) (Sigma-Aldrich, St. Louis MO, USA) [50] and 2% (*w*/*w*) gels [51,52]. PF is often used to quickly determine the depth of MN penetration since each layer is close to 100 microns [53]. We folded the PF into 8 layers, and we placed the MN on the top layer and pressed the MN with same force to press an elevator button. Next, we unfolded the PF sheets to examine the layers with holes on the microscope to visualize. If a hole was observed on the microscope, we considered the layer penetrated.

In addition, optical coherence tomography (OCT) was also used to visualize penetration studies of a 2% (*w*/*w*) gel since this mimics the properties of human skin [51,54]. Gel samples were imaged before and during MN penetration under OCT imaging for cross-sectional and volumetric visualization. A Telesto series OCT imaging system (TEL220PSC2, Thorlabs Inc., Newtown, NJ, USA) with an OCT-LK3 objective lens were used in the study. The system has a center wavelength of 1300 nm, and a bandwidth of 170 nm. The axial and lateral resolution of the system reported are 4.2 µm (in water) and 13 µm, respectively. Imaging data was acquired with 2D and 3D intensity mode at a A-scan rate of 48 kHz. From each sample, B-scans (4 mm, 1024 A-scans) and volumetric scans (4 mm × 2 mm area, 1024 A-scans by 200 B-scans) were collected. Post-processing of acquired images were performed with ThorImageOCT (version 5.4.9, Thorlabs Inc.). The cross-sectional imaging capability of OCT allows evaluation of the penetration depth and structural integrity of the MNs in the sample. We assumed a refractive index of 1.33. In this study, one layer of PF was placed on top of the 2% (*w*/*w*) gel to mimic the skin outermost layer, the stratum corneum. Two gels, ~250–300 µm thick were stacked to mimic epidermis layers [33] and another sheet of PF was placed in between the two gel layers to enhance needle visualization. Microneedles were placed on top of gel and pressed with the same force as used to press an elevator button.

### 2.5. Mechanical Tests

Compression tests are used to determine the modulus, elasticity, and malleability of a material. A compression test was conducted on three MNs patch in to determine the behavior of a PVA MN under a load using an Autograph AGS-J (Shimadzu, Kyoto, Japan). The patch was supported from below by a fixed, flat surface. A second flat surface, above and parallel to the supporting surface, was controlled to move in the direction of the lower surface, thereby compressing the material. The testing machine lowered the compression plate at a rate of 5 mm/min until the sample gave in. The following formulas were used to calculate the stress and strain of the PVA MN patch:(1)$$\sigma =F_{n}/A$$
where σ is the normal stress in N/mm^2^, $F_{n}$ is the normal force in N given by the testing machine, and *A* is the average cross section for the needles in mm^2^. This average cross section was calculated by taking the average of the base diameter (350 µm) and needle tip diameter (~10 µm). This average diameter was used to calculate the average cross section for a needle tip and multiplied to account for the 25 needles in the patch.
(2)$$\varepsilon =\Delta L/L_{{}^{\circ}}$$
where ε is the strain (unitless), ΔL is the change in height of the MN patch in mm, and $L_{{}^{\circ}}$ is the initial height of the microneedles (~600 µm).

### 2.6. Swell and Absorption Test

One method for capturing bacteria and bacterial DNA using PVA MN is to take advantage of the material’s hydrophilic and water absorption properties. Therefore, PVA liquid absorption is a critical property that directly relates to capture performance.

To qualitatively visualize liquid absorption performance, a 2% (*w*/*w*) gel was prepared. A small hole was made with the tip of a 25-gauge needle and 10 µL of green dye was injected into the hole. A sheet of PF was placed on top to ensure only the needle of the MN was extracting and not the underside of the MN patch. A MN was placed on top of the PF directly above the green dye spot as shown in Appendix A. The MN was applied for 10 min and then removed. A Kim wipe was used to blot excess of green dye off and the MN was examined for green color which indicated the MN absorbed the green dye.

To observe more detailed changes in the PVA MN swelling, scanning electron microscopy (SEM) was used to image a dry microneedle and a microneedle after 10 min with 10 µL of liquid. Each MN was sputter coated with gold and placed into the SEM Zeiss Merlin for imaging (Zeiss, Oberkochen, Germany).

In addition to these qualitative tests, a quantitative swell test was designed. To conduct a swell test, each individual MN patch was weighed while it was dry (*m_o_*) three times and then placed face down (needle side facing down) onto a 40 µL droplet of water and manually flattened briefly so that the needles were submerged in the water. After 10 min, each MN was removed from the water droplet with tweezers, and carefully placed on tissue paper for 10 s to remove the extra water sticking to the sides of the patch so that only the water absorbed within the MNs was left. Then, each wet MN patch (*m_t_*) was weighed three times. The percentage of swelling was determined by calculation from Equation (1).
(3)$$\%\;\mathrm{Swelling}=\left(\;\frac{m_{t}-m_{{}^{\circ}}}{m_{{}^{\circ}}}\;\right)\times 100\%$$

### 2.7. Time and Volume to Dissolve MN for PCR Testing

To enable direct PCR testing, it is critical to examine the volume and time needed to completely dissolve MN patches into a solution that can be pipetted accurately (a viscous solution is not pipetted accurately). The MN patches were inserted into 2% (*w*/*w*) gels for 10 min and then placed in a 1.5 mL Eppendorf tube. A volume of 15–100 µL of TE buffer was added to the tube. The tube was briefly vortexed and spun down before being placed in an incubator at 37 °C for 15–30 min or onto a heat block at 115 °C for 10 min. Pictures were taken after initial placement into Eppendorf tube and checked every 10 min. We considered the microneedle to be dissolved when you could not visually see any of the microneedles in the tube and liquid was easily pipetted.

### 2.8. Preparation of Pig Ear Skin and Penetration of MN in Porcine Ear Skin by OCT

Pig ears were obtained from Oak Hill Genetics from previously euthanized Yorkshire female pigs (32 kg). The samples were washed and soaked for 30 min prior to testing. Sterile surgical scissors were used to cut portions of the ear for testing.

OCT was used to image MN inserted into the skin of pig ears. OCT is a promising tool to image MN in tissue since OCT is nondestructive to tissues and tissue do not need pretreatment or chemicals for visualization [55]. In this test, we selected MN Design B (Table 1) due to penetration performance in prior tests. We tested the MN on a pig ear skin with hair and pig ear skin without hair to determine the effect of hair on the penetration performance. We assumed a refractive index was 1.33.

### 2.9. RNAse P Assay for Extraction Control

The RNAse P Assay detects the RNAse P gene which is often used as an endogenous control in sample specimens and is used to determine if a sample was adequate for testing DNA from clinical samples [56,57]. For reverse transcription, samples were held at 55 °C for 10 min, followed by cycling from 94 °C for 10 s to 60 °C for 10 s for 50 cycles using a Rotor Gene Q thermal cycle (Qiagen, Germantown, MD, USA). A forward primer of sequence 5′-GAT TTG GAC CTG CGA GCG-3′, a reverse primer of sequence 5′-AAG CGG CGT TCT CCA CGA-3′, and a probe sequence of 5′-Cy5-TTC CGA CCT-TAO-CAA GGC GCT GC-BHQ-3′. The Ct values from three experiments were compared in a one-way ANOVA test to assess sampling across the MN’s used for bacteria extraction from pig ears.

### 2.10. Determination of Bacteria Recovery Using qPCR

To determine the number of bacteria after reconstitution, qPCR was performed using the Luna^®^ Universal Probe qPCR Master Mix (NEB #m3004) according to manufacturer’s protocol. All primers, probes and targets were synthesized by Integrated DNA Technologies (Coralville, IA, USA). FlaB is a structural component of flagellin in spirochetes, and is expressed during infection in both ticks and hosts [58,59,60]. The synthetic flaB target consisted of 5′-CTC AAG CGT CCT GGA CTT TAA GAG TTC ATG TTG GAG CAA ACC AAG ATG AAG CTA TTG CTG TAA ATA TTT ATG CAG CTA ATG TTG CAA ATC TTT TCT CTG GTG AGG GAG CTC AAA CTG CTC AGG CTG CAC CGG TTC AAG AGG GTG TTC AAC AGG AAG GAG CTC AAC AGC CA-3′. We used IDT Primer Quest tool to generate 4 sets of primers specific to the synthetic flab target sequence. One set of primers was selected for further testing due to high reaction efficiency and single product melt analysis. A forward primer of sequence 5′-CAAGCGTCCTGGACTTTAAGA-3′, a reverse primer of sequence 5′-AGCTCCCTCACCAGAGAAA-3′, and a probe sequence of 5′-/56-FAM/TGTTGGAGC/ZEN/AAACCAAGATGAAGCT/3IABkFQ/-3′ were used. For the initial PCR step the samples were held at 95 °C for 2 min. The samples were cycled from 95 °C for 15 s, to 60 °C for 30 s and 72 °C for 30 s for a total of 45 cycles in a Rotor Gene Q thermal cycle (Qiagen, Germantown, MD, USA). In addition to a no template control, standards were made using 6 × 10^8^ copies/reaction of flaB target were diluted 10-fold sequentially to 6 × 10^1^ copies/reaction and 2 µL of each were added to each reaction for a total volume of 20 µL per tube. After the bacteria was reconstituted and diluted with water to achieve desired concentration. A total of 2 µL of these tubes were used in the PCR reaction in triplicate to determine the number of bacteria in the tube.

### 2.11. MN Application to Pig Ear Skin and Capture Test

To mimic a feeding pit, we designed a single cone that was 300 µm long and 200 µm diameter at the base. We pressed the 3D printed cone into the pig skin. Then, we pipetted 5–10 µL of low passage *B. burgdorferi* wild-type strain B31 clone 5A19 containing a range of bacteria into the feeding pit area intradermally. The PVA MNs were pressed onto both the clean and infected pig ear samples for 10 min. Afterwards, they were each removed and placed into a 2 mL Eppendorf tube with 50 µL of TE buffer followed by briefly vortexing the tube and incubating the tubes at 115 °C for 10 min. We assume the bacteria are lysed during this heat step. A positive control tube was prepared by pipetting 10 µL of the spirochete strain into a different tube with 100 µL of TE buffer. PCR was performed in triplicate for these samples to calculate the amount captured by the MN patches.

## 3. Results

### 3.1. Microneedle Designs

We tested various types of PVA and found that the higher molecular weight PVA would often gel and not dissolve into water as easily as other lower molecular weight PVAs. The 31–50 K PVA was selected because it easily dissolved in water and the MNs showed sufficient hardness when demolded from the negative silicone mold. Less than 10% PVA was not sturdy enough to form rigid needles. For this reason, 10% PVA was selected for our methods for further development.

For bacterial capture, it is critical to consider the depth the bacteria will be placed by the tick hypostome. Although the tick hypostome varies in length across species and stages, for the *Ixodes scapularis* tick, the maximum length is close to 500 μm. By imaging a tick hypostome, Figure 3A, we verified the length and approximated that the penetration was likely not beyond 50%, which suggests the bacteria are likely deposited close to a depth of 250–300 μm during the tick feeding. Assuming the penetration would be close to 50% of the MN length, we selected four microneedle designs to test (Table 1). Each microneedle design was placed on the side and imaged with a light microscope. The images were imported into ImageJ. Individual PVA microneedles were measured. The average length and standard deviation of the needles was reported for each design (Figure 3B).

3D printed microneedles were shorter (Appendix A) relative to the computer-aided design (CAD) file used to create them (Table 1). A loss of ~30% decrease in 3D microneedle height was found compared to the design, but the width maintained close approximation to the CAD file. This is a very similar 3D printing phenomena reported in other microneedle papers [61]. One explanation for this reduction in height but not width is due to the scattering of UV light during the exposure of larger portions of 3D prints compared to smaller size parts. The less material, the less light scatter to initiate photopolymerization to cure the material. Similarly, other authors found that with more light intensity, the more initiation of the photopolymerization. They also found an increase in size of structure increased with light intensity [62].

### 3.2. Penetration Test

The penetration of a MN to a specific depth is an important characteristic. The PF test results indicate that the shortest microneedle design (400 µm in length) is not able to penetrate to 300 µm. Additionally, the longest microneedle (850 µm in length on average) exhibited bending between PF layers and did not penetrate efficiently to the third layer. Designs B and C showed efficient penetration to the third layer of PF (Figure 4A).

With MN designs from Figure 3B, we inserted the MN into a gel and visualized penetration with OCT imaging. Results show Design A and D did not penetrate the gel efficiently (Figure 4B). The red false coloring (Adobe Photoshop) shows the empty space between the microneedle and the top layer of PF. Both Designs A and D show empty space and thus, reduced penetration while Designs B and C show much less empty space. Therefore, the results indicate the ideal length for 10% PVA microneedles appears to be between 500–750 µm in length. A OCT 2D cross-sectional (B-scan) of 2% (*w*/*w*) gel was imaged during application of a MN and after removal of a MN showing holes left behind in the gel (Appendix A).

### 3.3. Mechanical Tests

Young’s modulus was calculated to be approximately 0.0014 MPa. The yield strength for the patch is 0.00030 N/mm^2^. The deformation force was found to be 4.2 N. This is in the range of what others have found with similar PVA microneedles [63].

Light microscope images taken before and after weights were placed on MN patches. The MN could withstand up to 255 g of weight added before slight bending was observed. However, when 80 Kg was placed on the MN patch, the 90-degree bending was observed (Appendix A).

### 3.4. Swell and Absorption Tests

Light microscope images taken before and after insertion of Design B MN into a gel (Figure 5A). In Figure 5A, the left side shows a smaller microneedle compared to a more swollen microneedle. Additionally, a dry microneedle was imaged on SEM Figure 5B (left) compared to a SEM image of microneedle after 10 µL of liquid was added to the center and allowed to sit for 10 min (right). Beyond 10 min, the MN did not change in mass so 10 min was set as the saturation time point.

Microscope images show green dye in the needle tips indicating the microneedle absorbed the green dye (Figure 5C,D). The top view of the microneedle shows the green color was absorbed into the individual microneedle only (Figure 5E).

Results of the swell test using 40 µL of water show on average, the weight of the design B MN patches increased by approximately 46% after being placed on a water droplet (n = 4).

### 3.5. Volume Needed to Dissolve PVA MN

Four volumes were tested to dissolve a PVA MN (Figure 6). Testing showed the MN (mass ~11 mg) needed a volume 50–100 µL of TE buffer to fully dissolve. The 50 µL volume required incubation at 115 °C and vigorous vortexting. Although smaller volumes were tested, these solutions did not dissolve the MN after vortexing and 115 °C incubation (Figure 6). The results suggest dissolving the PVA MNs requires full submersion of the MN into the TE buffer.

### 3.6. Penetration of MN in Porcine Ear Skin by OCT

OCT was used to confirm the penetration of MN into porcine ear skin. Results show the MN successfully penetrated the porcine ear skin. However, for pig ear skin samples with hair (Figure 7A,B), there was a reduction in penetration as shown in Figure 7D. Red false coloring (Adobe Photoshop) was added to the space between the MN and the porcine skin to show the gap. In a pig skin sample with little to no hair (Figure 7F,G), the penetration was more successful. Results show greater penetration occurred in samples with no hair (Figure 7I,H) than in images of the samples with hair (Figure 7C,D). This suggest that before application of a microneedle, hair should be removed to enhance penetration depth of the MN.

### 3.7. RNAse P Assay for Sampling Control

A one-way ANOVA was used to compare the MN collection of pig samples by using the RNAse P assay. There was no significant difference found between collection of the MNs applied to a porcine ear regardless of bacteria present in testing (F = 0.25, *p* value = 0.79). The average Ct value of the applied MNs were found to be 27 with a standard deviation of 2 cycles (Figure 8). As a negative control, a MN was not applied to pig ear skin but directly dissolved and tested against the three groups. The dissolved MN showed a significant difference in Ct value compared to the other three groups (F = 214 and *p* value < 0.001).

All the MNs applied to the pig skin samples were positive as shown for both the positive MN and negative MN because each sample contained sufficient pig RNA sample. This test is for pig RNA detection in the MN so all MNs applied to the skin should be positive. In an assay that incorporates this control, if the MN sample is negative for RNAse P this indicates the MN did not collect sufficient sample to test and it is invalid.

### 3.8. MN Capture of Inactivated B. burgdorferi

For capture testing, we measured the extraction recovery of two concentrations of inactivated *B. burgdorferi* spirochetes with our MN patch placed on pig ear skin (Figure 9). A PCR was performed in triplicate for these samples to calculate the amount captured by the MN patches. The standard curve shows intensity of fluorescence versus cycle number with the separation of curves in proportion to the logarithmic dilutions of target (*B. burgdorferi* gene flaB) (Appendix A).

Performance of the MN patches to recover bacteria was obtained by comparing the concentration recovered by microneedle patches with the initial concentration of bacteria spiked into pig skin (positive control). Our results show an approximate 80% recovery of bacteria compared to the bacteria spiked. Variation was also seen in the PCR experiments with the microneedle’s recovery of bacteria (Appendix A). In other experiments, we found a large variation in the positive control tubes likely due to the bacteria clumping during the reconstitution and dilutions. For lower concentrations, variation was larger.

## 4. Discussion

In this report, PVA MN patches have been studied for their penetration performance, swelling characteristics and recovery of bacteria spiked into a pig ear. Our results showed the most successful MN was Design B, which was 500 µm in height (Table 1 and Figure 3B). Design B showed successful penetration beyond 200 µm (Figure 4 and Figure 5) which is the depth that bacteria would likely be found. Additionally, ISF fluid is also at depths > 200 µm so our MN show successful penetration into the area to collect LD biomarkers. Our results showed 46% swelling that can further be optimized with various other cross-linking reagent to enhance swelling [39]. After inserting into pig ear skin, the MN showed successful capture of inactivated *B. burgdorferi* (Figure 9). On average, our assay was able to capture approximately 80% of the bacteria spiked into porcine skin. Since the MN did not penetrate far enough to be painful (500 µm), the PVA MNs herein show promise as a detection tool that is easy to use and painless to apply.

We have developed a purification-free method to sample skin with a swellable MN that detects inactivated bacteria. Conventional tissue biopsy or tick testing takes hours by comparison and results in the destruction of the bacteria as well as a loss of starting material after numerous processing steps. We show the extracted bacteria can be directly detected by PCR without further processing steps (Figure 9). We also show 10 min is sufficient time to recover the majority of bacteria present. Further studies should confirm the optimal time to capture bacteria at different depths in the skin.

Our results show 50–100 µL of TE buffer is required to successfully dissolve the MN. Additionally, it is critical to vortex and incubate at 37–115 °C for the MN to fully dissolve prior to PCR testing. Future work may investigate the addition of smaller volumes of reagents that may disrupt the cross-linking of PVA and successfully dissolve the MN without diluting the sample. In this report, we did not test the stability of the microneedles before dissolving, however, researchers showed 10% (*w*/*v*) PVA MNs have been shown to be stable for up to a week after DNA extraction from plants [64]. This suggests that if PCR cannot be performed directly after extraction, non-dissolved MNs can be tested for up to a week after extraction. Given the ease of use and affordability, this bacteria extraction method could be further optimized to enable direct to lab processing for suspicious arthropod bites.

Human skin is the most suitable sample to evaluate microneedle performance, but the availability of human skin is limited by ethical, regulatory, and cost considerations [65]. Therefore, murine and porcine skin has been used as a substitute to human skin [66]. Porcine skin is considered a closer approximation to human skin since it is histologically and anatomically closer to human skin [67]. These similarities include the epidermal thickness, dermal-epidermal thickness ratio, hair follicle and blood vessel density and collagen and elastin content [65,68]. Specifically, porcine ear skin is 10 µm thick so it is closer to human epidermis thickness and therefore is considered a closer model to human skin [69]. Our comparison across the MN used in bacteria extraction showed each group was applied to the porcine skin long enough to capture pig RNA and that the groups did not differ in their extraction of pig RNA (Figure 8). In future work, a pig RNA control could be used in a multiplexed PCR reaction to verify the MNs recovered adequate sample regardless of bacteria present.

Although LD may present with the characteristic EM, other pathogens could still be transmitted with *B. burgdorferi* in the tick bite. These co-infections of multiple tick-borne diseases results in diagnostic challenges. Ticks may acquire and transmit many pathogens such as parasites, bacteria and viruses [70]. In 2018, one study examined 16,080 ticks that were mostly removed from humans and found that 88% were co-infected [71]. A MN-based method combined with multiplex PCR would give more information to the patient and physician about the possibility of co-infection of multiple tick-borne diseases.

There are several areas to further improve bacterial capture of this method. Although PVA does swell, other reports have shown much greater degree swelling (up to 450%) by using cross-linking reagents [39]. A greater amount of swelling would likely capture more bacteria and result in higher recovery. One other limitation in our approach is the use of inactivated bacteria. To match patient skin testing, live bacteria should be tested to better understand our assay performance and if additional steps for bacterial lysis are required prior to PCR. To increase live bacterial capture, one future approach is the use of a chemoattractant. Uptake of the spirochete in blood by an uninfected tick requires chemical signaling in the tick saliva to attract spirochete, which is the principle of xenodiagnoses (i.e., using a vector to acquire the pathogen for further testing) [72]. Recently, researchers have compared different motility assays to identify chemoattractants in tick saliva in order to improve diagnosis (via skin MN patch or subdermal injection/implant) [73,74]. Tick saliva contains many molecules that are able to be used by infectious agents to promote transmission [75]. A microneedle patch doped with chemoattractants for the bacteria may draw the bacteria to the patch for capture and detection while also blocking an inflammatory response to the MN [73]. Without the tick saliva, the host’s cells may induce an inflammatory response against the bacteria. Anti-inflammatory molecules in tick saliva prevent attraction of dendritic cells, mast cells, macrophages, and lymphocytes [76]. Thus, tick saliva increases the virulence and survival of *B. burgdorferi* by blocking the inflammatory response [77]. With the inclusion of chemoattractants and cross-linking agents to enhance swelling in a MN, the low levels of bacteria that remain problematic for traditional testing, may be overcome with such a technology.

The number of spirochetes in the skin can vary widely due to the stage of the tick attached and duration of attachment to host. We did not do a complete limit of detection in this report. We tested two concentrations of inactivated *B. burgdorferi* (Figure 9) based on what has been reported in skin biopsy reports [18], and one limitation in this report is that this concentration may not match the number from an initial tick bite. This needs further investigation and development. Our PCR was not reliable for detecting low amounts of the inactivated spirochetes. Attempts to test lower ranges were limited by variability of clumping of the fixed bacteria. Low detection of the bacteria could be overcome in a few ways. By picking a different gene target that repeats multiple times in the spirochete genome, a lower limit of detection could be achieved possibly. Nested PCR has shown a lower limit of detection of just one spirochete from culture [60]. Another way is digital PCR (dPCR) which allows individual segmentation of into wells with either none or one target before amplification. In 2020, researchers developed a dPCR assay that was able to detect three genome copies of *B. burgdorferi* [78]. By including an enrichment step to pre-amplify DNA prior to dPCR amplification, the authors overcame the low sensitivity in patient blood samples that are due to low numbers of bacteria [79].

## 5. Conclusions

This poly(vinyl alcohol) microneedle method enables rapid detection of tick-borne pathogens in skin models. We demonstrated the MN had sufficient stiffness to penetrate porcine ear skin and the MN could extract inactivated *Borrelia burgdorferi* for direct PCR detection. With further development, this technology could eventually achieve early detection of the causative agent of Lyme disease.

## Figures and Tables

**Figure 1 biosensors-12-00819-f001:**
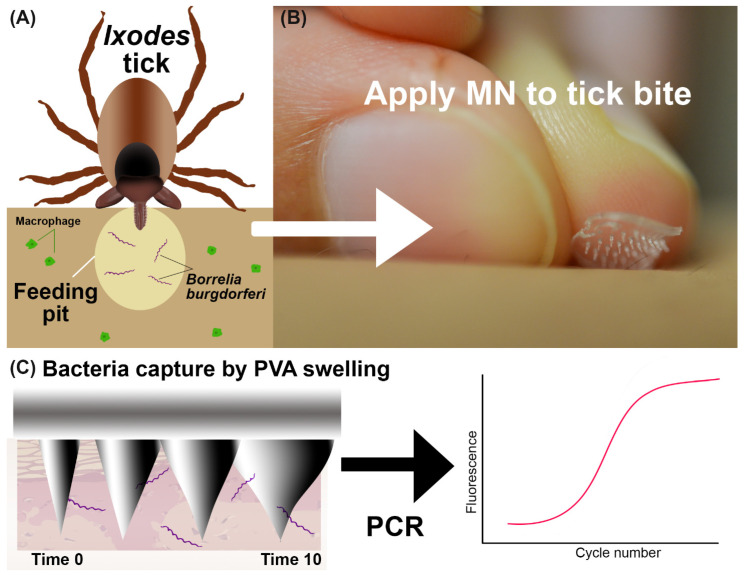
Overview of the bacteria detection method. (**A**) Graphic showing Ixodes tick feeding and inserting *B. burgdorferi* bacteria into skin. The tick saliva provides a feeding pit where bacteria are protected from immune cells. (**B**) A MN is placed onto the skin manually. (**C**) Illustration of MN capture of bacteria with swellable PVA and direct PCR detection.

**Figure 2 biosensors-12-00819-f002:**
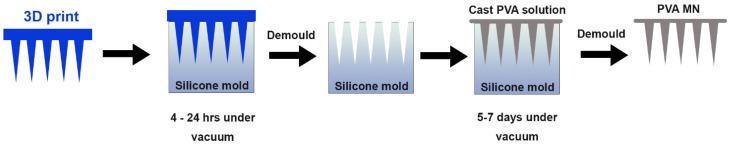
Fabrication method for PVA microneedle patches. A 3D print (blue) is used to cast the negative mold in silicone. In the final step, PVA (gray) is pipetted into the silicone mold and dried under vacuum for up to a week.

**Figure 3 biosensors-12-00819-f003:**
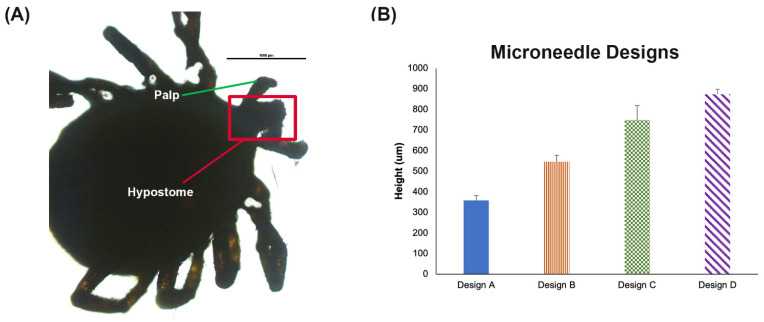
MN Design. (**A**) Light microscope image of a tick hypostome around 500 μm. Scale bar 1000 μm. (**B**) Measured height for each PVA MN design from Table 1 with four needles per patch (n = 3) ± SD.

**Figure 4 biosensors-12-00819-f004:**
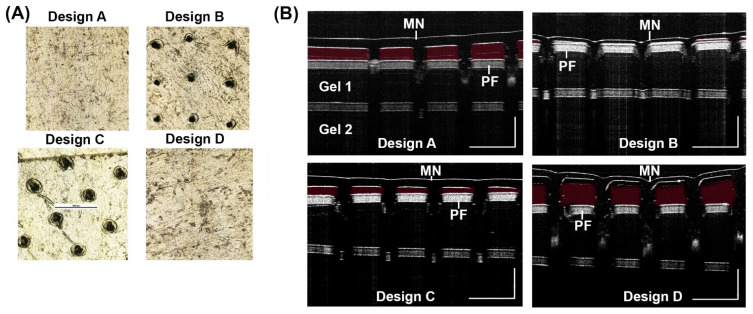
Penetration test for MN designs. (**A**) PF penetration to the 3rd layer was imaged on a light microscope for the four designs. (**B**) OCT 2D cross-sectional (B-scan) images were taken of each MN design after insertion into a 2% (*w*/*w*) gel with a one sheet of PF on top of the gel, a sheet of PF below gel 1, and another 2% (*w*/*w*) gel below the second sheet of PF (gel 2). Scale bar 1000 µm.

**Figure 5 biosensors-12-00819-f005:**
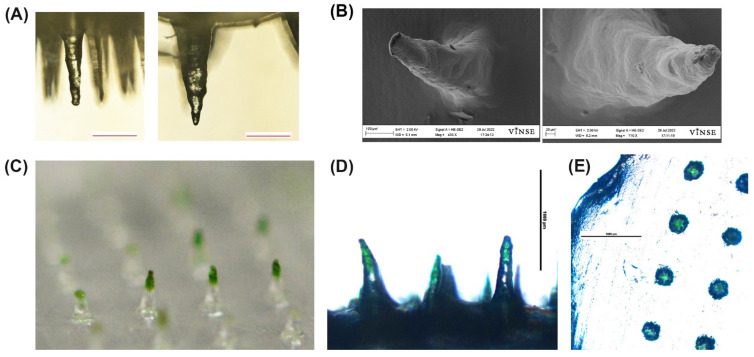
Representative images of the swell and dye absorption test. (**A**) Light microscope images of a MN before insertion into a gel (**left**) and after insertion (**right**). Scale bar 200 µm. (**B**) SEM images of a dry MN (**left**) and a MN insertion into liquid for 10 min (**right**). (**C**) Photograph (Nikon D610) of the microneedle patch after insertion into a gel with green dye below PF layer. (**D**) Light microscope image (4×) of microneedle after insertion into a gel with green dye below PF layer. Scale bar 1000 µm. (**E**) Light microscope image (4×) top view of the microneedle patch after insertion. Scale bar 1000 µm.

**Figure 6 biosensors-12-00819-f006:**
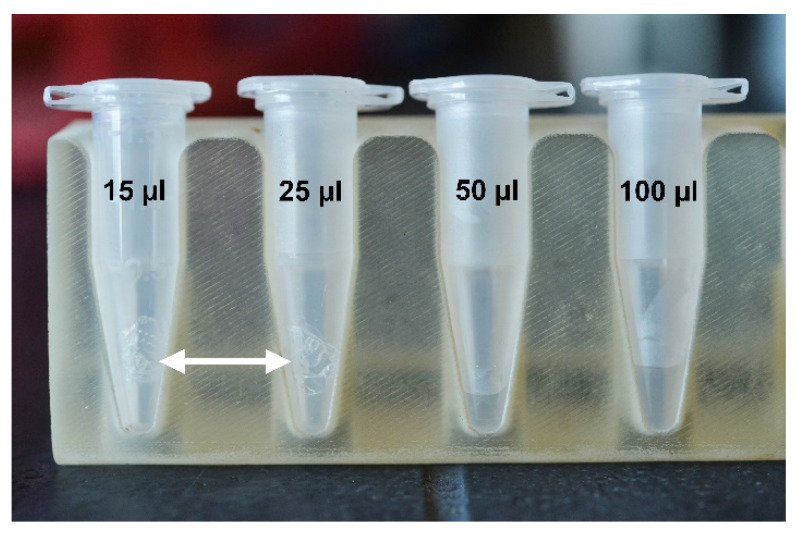
Representative images of microneedle inside tubes of TE buffer. A MN is placed into an Eppendorf tube with 15, 25, 50, and 100 µL of TE. (n = 3). White arrows indicate an undissolved MN in the tube after 10 min of incubation at 115 °C.

**Figure 7 biosensors-12-00819-f007:**
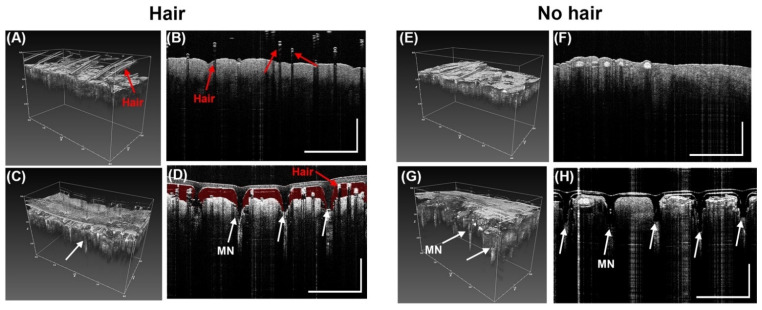
Representative OCT images from before and after penetrations of PVA MN array with microneedle height of 500 µm (Design B) into porcine ear skin. (**A**) Without a MN there are no holes in the skin with hair in 3D OCT image, (**B**) 2D cross-sectional (B-scan) image in skin sample with hair (**C**) MN inserted into sample with hair in 3D OCT image, (**D**) 2D cross-sectional (B-scan) image of MN inserted into skin sample with hair (red false coloring added to show gap between MN and skin). Scale bar 1000 µm. (**E**) Porcine skin with no hair in 3D OCT image before MN application. (**F**) 2D cross-sectional (B-scan) image in skin sample with no hair before MN application, (**G**) MN inserted into sample with no hair in 3D OCT image, (**H**) 2D cross-sectional (B-scan) image of MN inserted into skin sample with no hair. White arrows show the holes in the epidermis of the pig skin showing penetration (n = 3). Scale bar 1000 µm.

**Figure 8 biosensors-12-00819-f008:**
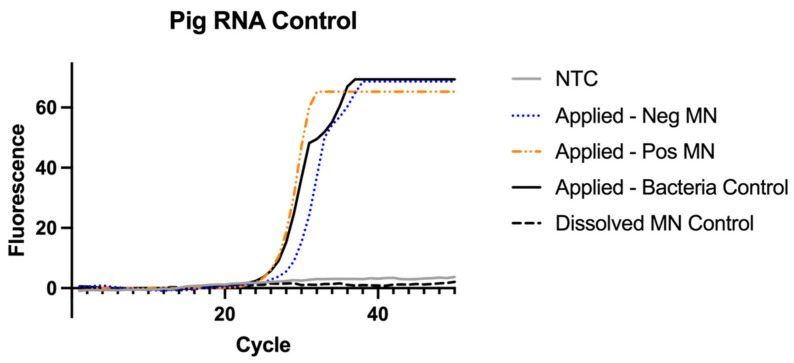
RNAse P Assay for extraction control results (n = 4). Samples from Figure 9 were re-run to test for the presence pig RNase P to ensure the MNs were applied long enough to the pig skin sample to collect RNA.

**Figure 9 biosensors-12-00819-f009:**
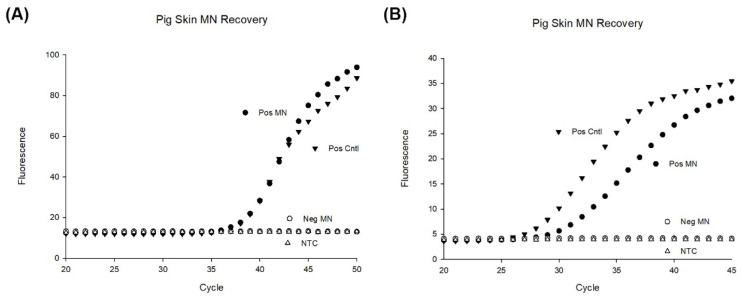
PCR results of MN patches applied to porcine ear skin. (**A**) 10^3^ bacteria were deposited to the depth of 300 µm and four MN were applied to the location for 10 min and subsequently dissolved in 50 µL of TE buffer for PCR (n = 4). A positive control was made by pipetting 10^3^ bacteria into 50 µL of TE buffer (n = 4). As a negative control, four microneedle patches were applied to porcine ear skin for 10 min and dissolved in 50 µL of TE buffer before PCR (n = 4). (**B**) 10^4^ bacteria were deposited to the depth of 300 µm and four MN were applied to the location for 10 min and subsequently dissolved in 50 µL of TE buffer for PCR (n = 4). A positive control was made by pipetting 10^4^ bacteria into 50 µL of TE buffer (n = 4). As a negative control, four microneedle patches were applied to porcine ear skin for 10 min and dissolved in 50 µL of TE buffer before PCR (n = 4).

**Table 1 biosensors-12-00819-t001:** Parameters of the 3D printed 5 × 5 microneedle cast prints used for molding.

MN Patch ID	Needle Length (µm)	Base (µm)	Distance between Needles (µm)
Design A	1000	250	1000
Design B	1000	350	1000
Design C	1200	350	1000
Design D	1600	350	1000

## Data Availability

All data available in manuscript and Appendix A.

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
