# Peer review of "Direct Capture and Early Detection of Lyme Disease Spirochete in Skin with a Microneedle Patch"

_biosensors, 2022, doi:10.3390/bios12100819_

Round 1

Reviewer 1 Report

The manuscript aims at providing a more sensitive approach to early Lyme disease diagnosis through the use of a microneedle device for detection of the causing agent for Lyme disease. The current standard set by the CDC (ELISA) is limited in sensitivity in large part due to the sample being used (blood) and the delay in development or lack of an immune response by the patient. The authors provide a novel, low-cost approach for obtaining a sample directly from the infection site and testing directly for the  bacterial pathogen. The method requires fabrication of a low cost polymer microneedle and direct application to the skin to retrieve the sample. The device is dissolved in water and analyzed by downstream PCR. This approach improves the sensitivity by detecting the bacteria directly from the lesion site. It is also a less invasive procedure than the typical blood draw required for ELISA.

General Comments

The manuscript is clear to follow and includes an appropriate introduction, describes all methodology used and the results support the hypothesis. I did not find any ethical violations either. Therefore, I recommend acceptance with minor revisions.

Specific Comments

69 I suggest changing bacteria DNA to bacterial DNA

82 This source discusses the potentially inhibitory effects of tissue fixatives such as ethanol or paraffin. You may want to clarify the inhibition is due to commonly used fixatives and not substances from the fresh biopsy.

142-144 Is the spirochete hypostome shape relevant to needle shape? Are the needles intended to fit into an existing cone or puncture their own? I believe approximate length is relevant to ensure the sample is collected from the right stratum, but is the shape/size equivalency relevant beyond this understanding? If so, can you elaborate on the reasoning? Otherwise, you may consider moving to introduction.

258. This reference is very old and refers to detection using a Western Blot. Is there a more recent reference that supports the use of PCR for detection of the FlaB gene? Was the primer sequence designed specifically for this project or is it from another published manuscript? If it is from another manuscript, that can be used as a reference.

269 For the synthetic control, did you add 2 ul of 6x10^1 dilution to each PCR? What is the total PCR volume for each reaction?

271 Was any kind of digestion or lysis performed on the bacteria? Are the 2 ul used in the PCR the same TE buffer with bacteria described in lines 280-283?

365 Why were the smaller volumes not able to be pipetted into PCR? Is this the high viscosity problem mentioned in 234-235? Can they be diluted by a small factor without affecting sensitivity?

411 Figure 9. Was this experiment done more than once? In addition to this single curve for MN and control, an average and variability metric might be interesting to the reader and further support the reproducibility of the experiment.

433 Does this imply that all DNA is released by thermal lysis during PCR cycling without the need for a separate digestion process?

450-452 Since your control is RNA based, you may want to clarify this is a pig sample nucleic acid control, particularly to avoid confusion with line 452 where a DNA test is proposed.

455 Since you have defined EM as erythema migrans you can use the abbreviation

489 Were other target concentrations tested? In real-world conditions, the bacterial load may vary and it is important to establish a working range for the assay. If you have experimental data, even if limited, it may be worthwhile to include in the manuscript as supplementary data.

501 Abbreviated EM can be used here as well

Reviewer 2 Report

In their manuscript "Direct Capture and Early Detection of Lyme Disease Spirochete in Skin with a Microneedle Patch", the authors describe a PVA microneedle (MN) tool fabricated using 3D printed microneedle molds for sampling interstitial fluid (ISF) and capturing spirochetes directly from skin. Using Parafilm, 2% (w/w) gel and pig ear skin, the fabricated microneedles were examined for penetrating the stratum corneum of the skin for extraction of ISF. The swelling and dissolving properties of the PVA microneedles were investigated and optimized for capturing B. burgdorferi, followed by systemic spirochetes quantification by conducting PCR after dissolving PVA microneedles. Overall, the authors present an impressive amount of work, including effort at successful capturing and detecting B. burgdorferi from skin ISF.

This study would be of interest to the general Biosensors readership. However, the manuscript suffers from several issues (detailed below). With further clarification or improvement, the reviewer believes this study can be considered for this journal.

1. Line 97 – All the abbreviations should be given the description for the 1st presence in each section.

2. Line 150 – From the Table 1, the needle pitch is 1000 µm in all types of designs. It would be great if authors can explain more details for this design. Is there any special considerations for this needle pitch? If not, is there any chance for missing inspection because of the relatively large needle pitch?

3. Line 154 – Different types of PVA were tested ranging from 13-124K molecular weight, but without showing the data of different microneedle patches. Can the author provide the data for selecting PVA with 30–50K molecular?

4. How about the mechanical properties of the PVA microneedles? It is suggested to provide the force-travel curves of microneedles using micro-mechanical test machine.

5. For future clinical applications, how long the fabricated solid PVA microneedles can be stored?

6. In the Materials and Methods. It’s better to provide the statistics.

7. In all images used in figures, scale bar should be provided for readers. For example, there is no scale bar in Figure 4a and Figure 5C.

8. In all figure captions, please use the consistent font format.

9. In Figure 9, please provide more descriptions for ‘Pos MN’and‘Pos Cntl’.

10. Please further condense the conclusions.

11. The reference format should be consistent in different references such as p. 781 in ref. 1 and pp. 750–764 in ref. 5.

Author Response

Please see the attachment, thank you!

Round 2

Reviewer 2 Report

I am satisfied with the authors' responses to my comments.  I would be pleased to recommend the acceptance of the manuscript in the present form.